# Evaluating the impact of curfews and other measures on SARS-CoV-2 transmission in French Guiana

Alessio Andronico [1,11 ✉], Cécile Tran Kiem [1,2,11], Juliette Paireau [1,3], Tiphanie Succo[4], Paolo Bosetti[1], Noémie Lefrancq [1], Mathieu Nacher[5,6], Félix Djossou[7], Alice Sanna[8], Claude Flamand [1,9], Henrik Salje [1,10], Cyril Rousseau[4] & Simon Cauchemez [1]

While general lockdowns have proven effective to control SARS-CoV-2 epidemics, they come with enormous costs for society. It is therefore essential to identify control strategies with lower social and economic impact. Here, we report and evaluate the control strategy implemented during a large SARS-CoV-2 epidemic in June–July 2020 in French Guiana that relied on curfews, targeted lockdowns, and other measures. We find that the combination of these interventions coincided with a reduction in the basic reproduction number of SARS-CoV-2 from 1.7 to 1.1, which was sufficient to avoid hospital saturation. We estimate that thanks to the young demographics, the risk of hospitalisation following infection was 0.3 times that of metropolitan France and that about 20% of the population was infected by July. Our model projections are consistent with a recent seroprevalence study. The study show-cases how mathematical modelling can be used to support healthcare planning in a context of high uncertainty.

[1] Mathematical Modelling of Infectious Diseases Unit, Institut Pasteur, UMR2000, CNRS, Paris, France. [2] Collège Doctoral, Sorbonne Université, Paris, France. [3] Santé Publique France, French National Public Health Agency, Saint Maurice, France. [4] Santé Publique France Guyane, French National Public Health Agency, Cayenne, France. [5] Centre d'Investigation Clinique Antilles Guyane, CIC INSERM 1424, Centre Hospitalier Andrée Rosemon, Cayenne, France. [6] DFR Santé, Université de Guyane, Cayenne, France. [7] Service des Maladies Infectieuses et Tropicales, Centre Hospitalier de Cayenne, Cayenne, France. [8] Agence Régionale de Santé de Guyane, Cayenne, France. [9] Epidemiology unit, Institut Pasteur in French Guiana, Cayenne, France. [10] Department of Genetics, University of Cambridge, Cambridge, UK. [11] These authors contributed equally: Alessio Andronico, Cécile Tran Kiem. ✉email: alessio.andronico@pasteur.fr

Following its first detection in China in December 2019, the SARS-CoV-2 virus has quickly spread around the world[1]. To avoid saturation of their healthcare systems, many countries enforced nationwide lockdowns. While such an approach has demonstrated its efficacy for transmission control[2–5], it comes with very high social and economic costs[6,7]. As a consequence, lockdowns cannot be sustained for long periods of time and it remains essential to identify sets of interventions with lower impact on society that are effective enough to contain epidemic rebounds of SARS-CoV-2.

While mathematical modelling can help evaluate the likely impact of different strategies, demonstration of efficacy comes when these approaches are successfully implemented in the field. We therefore critically need to determine from local experiences of epidemic management which set of interventions may be sufficient for the control of a SARS-CoV-2 epidemic while having the lowest societal cost. Here, we report on the local experience of French Guiana, a French overseas territory located in Latin America in the Amazonian forest complex, where authorities managed to contain a large SARS-CoV-2 epidemic with the use of curfews, local lockdowns, and other measures. We describe the epidemic dynamics, the interventions that were implemented and use a mathematical model to evaluate how the strategy that heavily relied on curfews impacted SARS-CoV-2 spread. Second, French Guiana has similar healthcare and surveillance as Metropolitan France but a much younger population. The comparison of SARS-CoV-2 epidemics in these two locations therefore offers an interesting setting to disentangle the impact of demographics on COVID-19 burden from that of other variables that typically also vary in international comparisons (e.g., healthcare structure, surveillance…). Finally, we show how mathematical modelling was used throughout the outbreak to support healthcare planning and policy making in a context of high uncertainty.

## Results

**Epidemic of SARS-CoV-2 in a young population: impact on healthcare demand.** The severity of SARS-CoV-2 infection increases with the age of the individual[2,8,9]. As a consequence, the impact of a SARS-CoV-2 epidemic on the healthcare system is expected to vary with the demographic structure of the population it is spreading in[10]. The population of French Guiana is substantially younger than that in metropolitan France, with a median age of 27y in French Guiana compared to 42y in metropolitan France (Fig. 1b). We used our mathematical model[2] to anticipate how these differences were expected to affect stress on the healthcare system for a given level of circulation of the virus.

During the first pandemic wave in metropolitan France, we estimated age-specific probabilities of hospitalisation given infection under the assumption that children were half as infectious as adults (Fig. 1c)[2]. We found that, on average, in metropolitan France, a person infected by SARS-CoV-2 had 3.5% [1.9%, 5.8%] probability of being hospitalized (Fig. 1d)[2]. Applying these age-specific probabilities to the demographic structure and expected contact patterns in French Guiana, we anticipated that the average probability of hospitalization upon infection would be 1.1% [0.6%, 1.8%] in this population (Fig. 1d). This means that, for the same number of infections, we expected 0.32 times as many hospitalisations in French Guiana as in metropolitan France.

**Measures implemented to control the spread of SARS-CoV-2 in French Guiana.** Like the rest of France, a territory-wide lockdown was imposed in French Guiana from March 17th 2020 to May 11th 2020. At that point, the lockdown was eased but not entirely lifted: schools, places of worship and movie theatres stayed closed, while restaurants and bars were allowed to reopen but were limited to outdoor sitting for on-site dining. A curfew was established from 11PM to 5AM every day except in Saint Georges, a city located on the border with Brazil, where a complete lockdown was maintained in order to contain the number of imported cases from the neighbouring country. Terrestrial borders were closed and travel restrictions were implemented to reduce the risk of spatial spread. Despite these measures, and concomitantly with the acceleration of the epidemic in Brazil (that neighbours French Guiana), the number of cases started to rise at the end of May 2020. In response to this rise, control measures were strengthened on June 10th, with the general curfew being extended from 9PM to 5AM during weekdays and for the entire day on Sundays. On June 18th, the curfew was extended again from 7PM to 5AM during weekdays and for the entire weekend starting on Saturday 3PM. From June 25th, additional measures included the start of the curfew at 5PM during weekdays and at 1PM on Saturday for the rest of the weekend, enforced closure of all restaurants, the closure of the Brazilian border and the lockdown of 23 high-risk areas. There were also important screening campaigns with more than 1300 tests per 100 000 inhabitants per week in the weeks following June 15th 2020. A partial easing of the more stringent measures - in particular the local lockdowns - took place on July 25th (Fig. 1e). Overall, the implementation and schedule of the curfews were adapted to each area according to the epidemiological situation: the tightening (or easing) of the curfews and the closure of nonessential businesses were all decided at the municipality level - and without resorting to a fixed threshold - by monitoring the number of detected and hospitalized cases. The measures were taken following the progression of the epidemic, which started in the east, and it then progressed along the coast (Cayenne and Kourou) to finally reach the west of the region (St-Laurent-du-Maroni) (see Fig. 1).

**Planning for a pessimistic outcome.** A key challenge for the management of such an epidemic is that it is not possible to evaluate the impact of new control measures on hospital admissions for the 2-3 weeks that follow their implementation (about 11 days from infection to hospitalisation[11] and between 5 and 10 days to accumulate sufficient data to characterize trends post intervention). At a time of such high uncertainty, it is important to plan adequate healthcare capacity and a potential strengthening of control measures in case transmission rates following intervention are not sufficiently reduced. We ran two analyses during this period to support such planning.

**What if control measures do not reduce transmission?** To support healthcare planning and ensure appropriate scaling of the local ICU capacity, we evaluated healthcare demand in a pessimistic scenario wherein local transmission rates would remain unchanged despite the additional control measures. This was done considering a broad spectrum of scenarios for the probability of hospitalisation of an infected individual (baseline: $p_H = 1.1\%$; low: $p_H = 0.6\%$; high: $p_H = 1.8\%$) (Fig. 1d) and the duration of stay in ICU (baseline: $\tau_{ICU} = 11.4$ days; short: $\tau_{ICU} = 8.0$ days; long: $\tau_{ICU} = 15.0$ days). Both estimates of the probability of ICU given hospitalization and of the average duration of stay in ICU were obtained using the model described in[12] (see also the Supplementary Information and Supplementary Table 1).

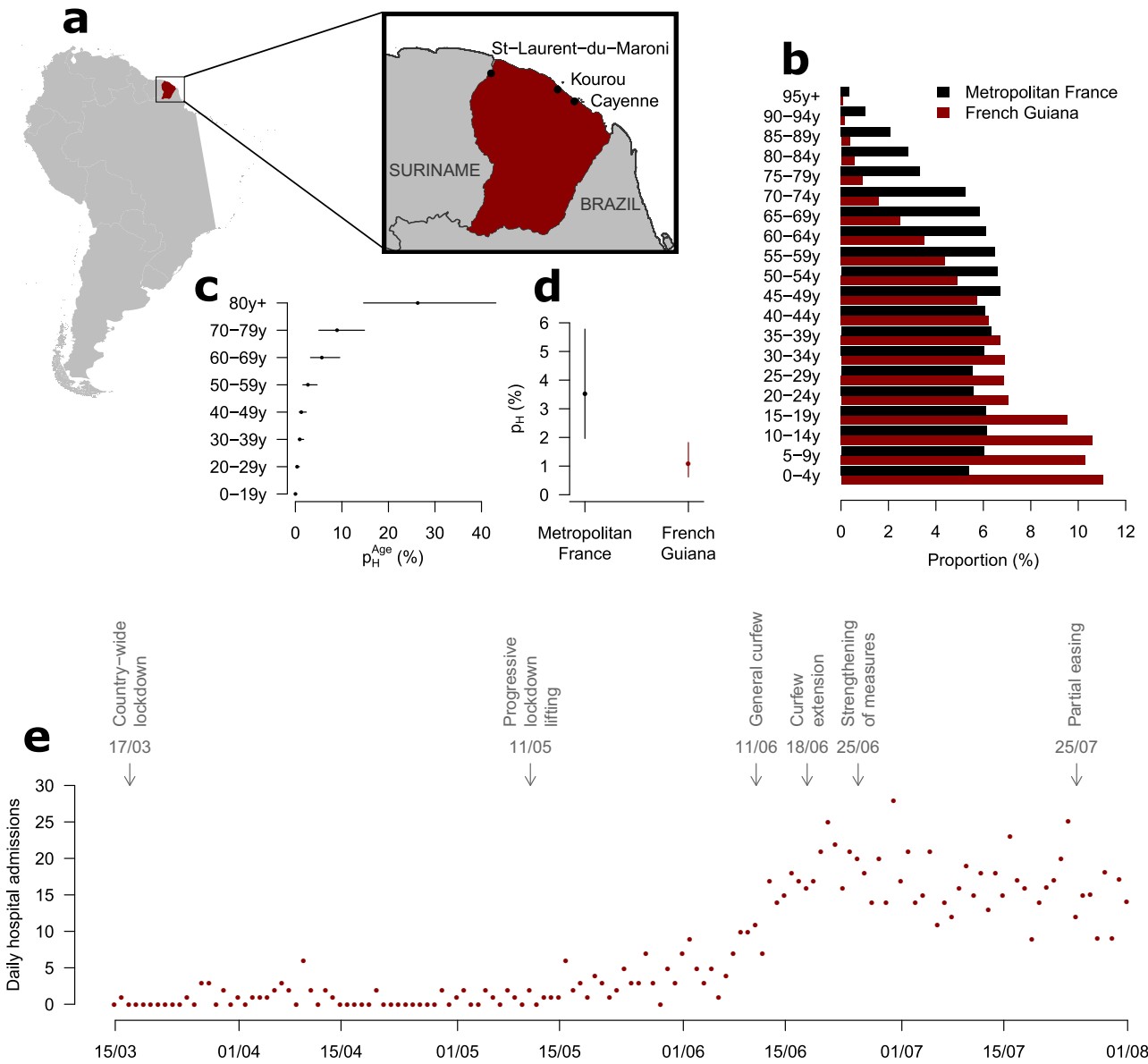

**Fig. 1 Geography, demographics, and timeline of interventions. a** Map of French Guiana (Source: https://gadm.org/maps.html). **b** Population pyramids for Metropolitan France and French Guiana. **c** Age-specific probability of hospitalization given infection $p_H^{Age}$ in Metropolitan France (%). **d** Average probability of hospitalization given infection $p_H$ in Metropolitan France and French Guiana (%). **e** Daily hospital admissions in French Guiana and timeline of interventions. In panels **c** and **d**, dots denote posterior means while bars denote 95% credible intervals.

Calibrated to data available on June 18th, our model (model M1) identified that there had been an acceleration of the epidemic around May 20th, with the basic reproduction number increasing from 1.35 [1.26, 1.45] (before May 20th) to 1.78 [1.68, 1.88] (afterwards). In the scenario where the transmission rate remained unchanged following interventions, the peak of the epidemic was expected in July (Fig. 2a, b). Depending on the severity scenario, the peak number of daily hospitalisations was projected at 48 [34,65] for average severity (low severity: 28 [18,40]; high severity: 75 [55,96]) while the peak number of daily ICU admissions was projected at 11 [4,18] (low severity: 6 [2,11]; high severity: 16 [8,25]). The number of general ward beds required at the peak was estimated at 454 [383, 528] for average severity (low severity: 262 [221, 305]; high severity 715 [592, 831]), and the number of ICU beds at 110 [86, 137] (low severity: 63, [47,82]; high severity: 173 [135, 210]). Supplementary Fig. 1 shows that the projected timing

and intensity of the peak were overall more sensitive to the probability of hospitalisation $p_H$ than to the duration of stay in ICU $\tau_{ICU}$.

**Impact of a short lockdown on peak intensity.** Given the quickly expanding epidemic, the limited ICU capacity and the possibility that existing control measures might not sufficiently reduce transmission, the option of imposing a short lockdown in French Guiana was also considered by French authorities. We assessed how such a lockdown might ease peak healthcare demand, assuming that it would lead to similar transmission rates as those estimated in metropolitan France with a lockdown reproduction number of 0.7. This was done considering different start dates and durations for the lockdown (Fig. 2c, d). We found that a territory-wide 10 days lockdown would reduce the required number of general ward beds from 454 [383, 528] to 256 [222,

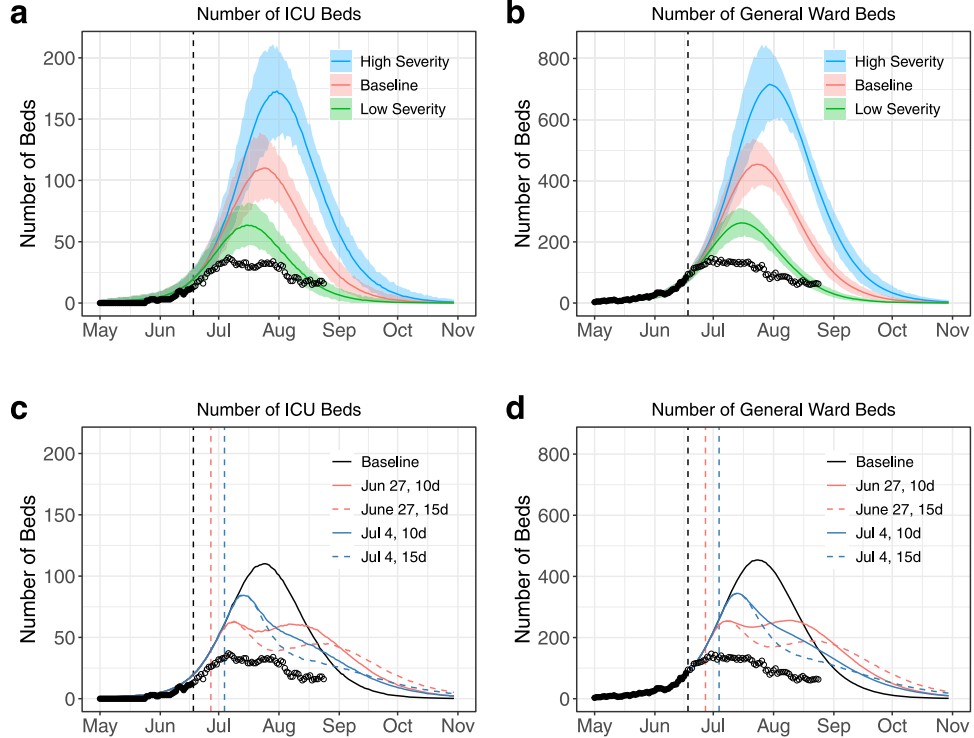

**Fig. 2 Analyses made on June 18th 2020 describing a scenario with no change in transmission rates and the impact of a short-term lockdown.** **a**, **b** Projections for the number of ICU and general ward beds required under different severity scenarios (baseline in red, low severity in green, high severity in blue). Solid lines indicate model posterior means while colour areas indicate 95% credible intervals. **c** and **d** Projections for the average number of ICU and general ward beds required under different territory-wide lockdown scenarios (black represents our baseline model, red represents lockdowns starting on June 27th, blue represents lockdowns starting on July 4th, solid lines correspond to lockdowns lasting for 10 days, while dashed lines correspond to lockdowns lasting for 15 days). In all panels, black dots indicate data used to calibrate the models, while empty circles denote data not available at the time of the analyses. The black dashed line in all panels indicates the date of the analyses (June 18th). The coloured dashed lines in panels **c** and **d** indicate the start of the simulated lockdown (red for June 27th and blue for July 4th).

290] (if started on June 27th) or to 345 [276, 427] (if started on July 4th). Similarly, the expected number of required ICU beds was projected to decrease from 110 [86, 137] to 63 [45,86] (lockdown starting on June 27th) or to 84 [61, 111] (lockdown starting on July 4th).

Eventually, given the societal and economic cost associated with a territory-wide lockdown, this strategy was ruled out and it was decided to implement less drastic measures accompanied by an increase of ICU bed capacity and the planning of patient transfers to hospitals in Martinique and Guadeloupe.

**Initial estimates of the impact of interventions on the epidemic trajectory**. We ascertained the impact of the new interventions by adding a change point for the transmission rate (model M2): the timing of the additional change point was estimated by comparing the models' DICs (see Methods). From June 27th onwards, this model had better DIC support than model M1 with no change in transmission (Supplementary Fig. 2). Model M2 estimated that the transmission rate was reduced from June 15th [10th, 19th], coincidentally with the strengthening of control measures (Supplementary Fig. 3). According to the model, the basic reproduction number went from 1.40 [1.32, 1.49] before May 20th to 1.71 [1.65, 1.77] between May 20th and June 15th and 1.14 [0.95, 1.31] after June 15th (Fig. 3a). This suggests that the strict curfew measures were successful at reducing transmission. With these reduced transmission rates, projections of the number of hospital and ICU beds required at the peak dropped to

28 [17,42] ICU beds and 162 [127, 203] general ward beds (Fig. 3c, d, Supplementary Fig. 4).

**Latest assessments, model validation and improvements**. We retrospectively validated the model by comparing its estimates to the results of a seroprevalence survey performed between July 15th and July 23rd with the Euroimmun assay[13]. Assuming a time-dependent assay sensitivity (0% during incubation, 30.3% up to 10 days after symptom onset, 75% between 10 and 20 days after symptom onset, and 93.8% afterwards) as per the distributor specification, our model estimated that 17.6% [17.2%, 18.0%] of the population was seropositive for SARS-CoV-2 between July 15th and July 23rd for average severity (low severity: 31.8% [31.1%, 32.6%]; high severity: 10.7% [10.5%, 11.0%]) (Fig. 4a), with model M2 calibrated using data available on 25 August 2020 (Supplementary Table 1). Estimates for average severity are close to the seroprevalence of 15.4% [9.3%, 24.4%] obtained in the serosurvey, indicating that our average severity scenario remains the one that is best supported by the data. Projecting forward, and contingent on control measures being maintained, we anticipated that 30.6% [29.9%, 31.3%] (low severity: 51.9% [50.9%, 52.9%]; high severity: 19.3% [18.8%, 19.7%]) of the population in French Guiana would have been infected by 1 October 2020 (Fig. 4b).

Figure 4c compares the age distribution of hospitalized cases in Metropolitan France and in French Guiana and admitted to a hospital between March 1st and August 25th (see also Supplementary Fig. 5 for the per-capita hospitalizations). To highlight how the different population age structure affects

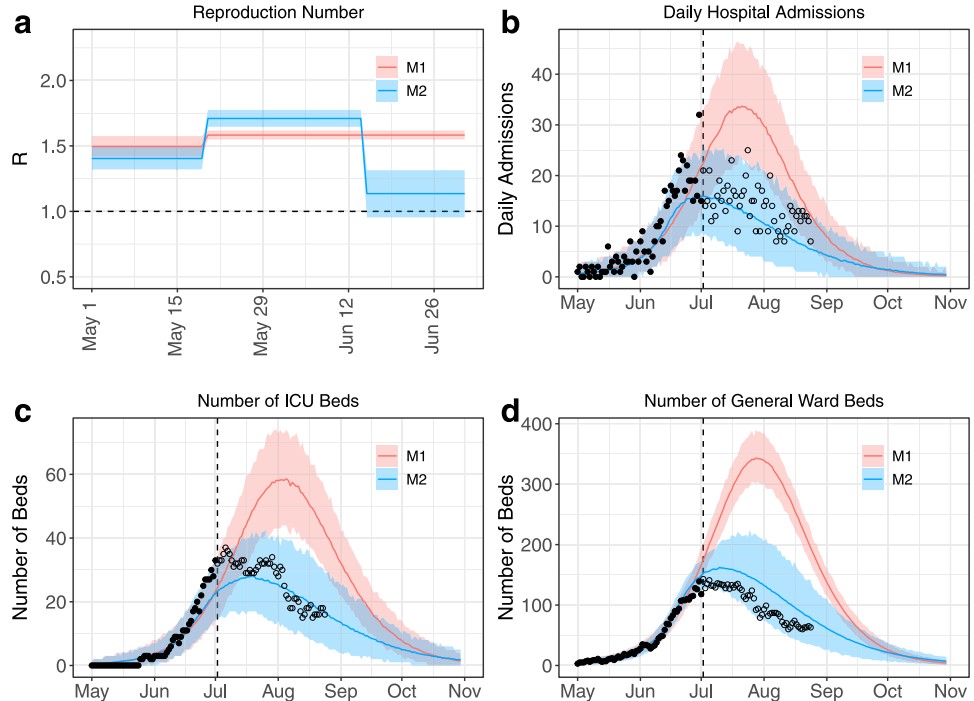

**Fig. 3 Analyses made on July 2nd 2020 evaluating the impact of control measures implemented in French Guiana on transmission and healthcare demand. a** Estimated reproduction number through time. The horizontal dashed line indicates a reproduction number R = 1. **b, d** Projections for the number of daily hospital admissions (**b**) and ICU (**c**) and general ward (**d**) beds. Solid lines indicate model posterior means while colour areas indicate 95% credible intervals. Red is used for model M1 (one change point for the transmission rate), while blue is used for model M2 (two change points for the transmission rate). In all panels black dots indicate data used to calibrate the models, while empty circles denote data not available at the time of the analyses. The dashed line panels **b–d** indicates the date of the analyses (July 2nd).

hospital admissions, in Fig. 4d we show an epidemic with the same trends in daily number of infections as in French Guiana but with the population demographics of the Cher department. This department has roughly the same population size of French Guiana (296,404 individuals for Cher vs 290,691 for French Guiana) but is located in Metropolitan France. To do this, we used model M2 and set all its parameters to the values estimated for the French Guiana epidemic with the exception of the probability of hospitalization given infection, for which we used the value estimated for Metropolitan France ($p_H = 3.5\%$). As expected, the peak would have been about 3 times greater, with important consequences in terms of hospital overload and epidemic management.

## Discussion

In this paper, we characterized the epidemic dynamics of SARS-CoV-2 in French Guiana, evaluated the impact of control measures that were implemented to contain a large SARS-CoV-2 epidemic there, and described how mathematical modelling was used during this crisis to support policy making and planning.

The nation-wide lockdown that was implemented across France from March 17th 2020 to May 11th 2020 likely prevented a surge of SARS-CoV-2 infections in French Guiana during this period. However, while a number of control measures remained in place in French Guiana after the lockdown, they were insufficient to stop an important epidemic rebound, which coincided with the surge of cases observed in Brazil, a country that has experienced a very important pandemic wave[14,15], most notably in the Amazonian states. Confronted with an important surge in COVID-19 cases, French authorities implemented a set of strong measures including curfews and localized lockdowns. During

curfews, individuals can go to work and live a relatively normal life during the day, but social interactions are limited in the evenings and weekends. This approach therefore targets social interactions among family members, friends or close acquaintances, where social distancing is likely to be more lax. While smaller than that of a full lockdown, the economic impact of a curfew remains important in particular for the hospitality, catering and recreational sectors, as well as for a large part of the undeclared jobs on which the most precarious rely on in French Guiana.

We estimate that, added to existing measures, these interventions coincided with a reduction of the basic reproduction number by 36% from 1.7 (prior to interventions) to 1.1 (following implementation). This change in epidemic dynamics strongly reduced predicted ICU beds needs for the epidemic peak from 110 to 32, thereby avoiding saturation of ICUs. The territory was also able to manage the influx of patients thanks to an expansion of ICU capacity (from 11 on May 1 2020 to 54 on July 22 2020) and the transfer of 7 ICU patients to Martinique and 6 to Guadeloupe, two French overseas territories located in the Caribbean. While we can correlate estimates of the reproduction number with the timing of interventions, it is not possible from such analysis to demonstrate causality or to estimate the relative contribution of the different measures that were implemented at the time.

In agreement with a seroprevalence study[13], we find that the infection attack rate of SARS-CoV-2 in French Guiana was one of the highest in France after the first wave, likely higher than that estimated for Grand Est (7.7–10.2% between May 4th and June 22nd) and Île-de-France (Paris area) (9.1–10.9% between 4 May and 22 June), the two regions of metropolitan France that have been the most affected by the first pandemic wave[16]. This may

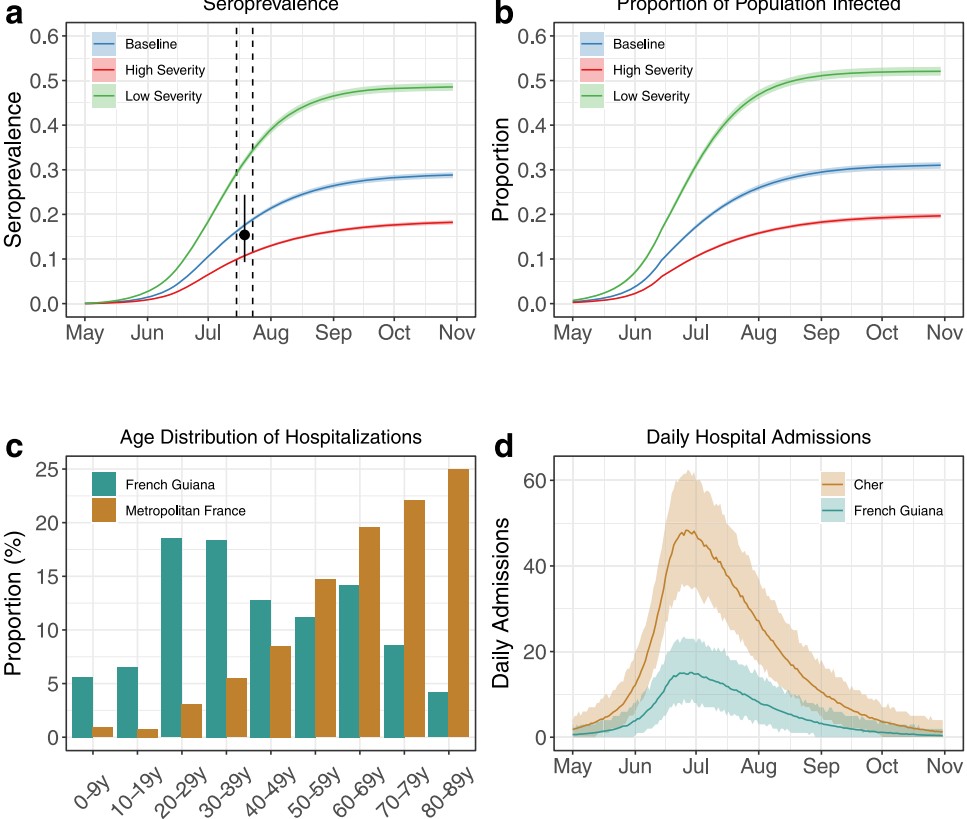

**Fig. 4 Analyses made on 25 August 2020. a** Projections for the seroprevalence measured with the Euroimmun assay. Solid lines indicate model posterior means while colour areas indicate 95% credible intervals. The black dot and vertical bar indicate seroprevalence estimates from[13] between 15 and 23 July 2020 (dashed lines). **b** Projections for the proportion infected. Solid lines indicate model posterior means while colour areas indicate 95% credible intervals. **c** Age distribution of hospitalized cases in Metropolitan France (brown) and French Guiana (green). **d** Simulated numbers of hospital admissions in the Cher department (Metropolitan France, brown) compared to those obtained for French Guiana (green). Solid lines indicate model posterior means while colour areas indicate 95% credible intervals.

seem surprising since the impact of the epidemic on hospitalisations and deaths was substantially lower in French Guiana (183 hospitalisations by July 1st and 13 deaths per 100,000 inhabitants by July 18th) than in Grand Est (276 hospitalisations by May 25th and 60 deaths per 100,000 inhabitants by June 11th) and Île-de-France (280 hospitalisations by May 25th and 56 deaths per 100,000 inhabitants by June 11th). This apparent discrepancy was anticipated by our model and can be explained because the population of French Guiana is substantially younger than that of metropolitan France (Fig. 1). This shows that, as previously documented[2,8,9], it is essential to account for the age structure of a population to properly evaluate the impact of SARS-CoV-2 on its healthcare system. In an older population, it is likely that pressure on the healthcare system would have occurred earlier in the epidemic, leading to earlier implementation of control measures and lower seroprevalence. Improvements in patient management thanks for example to anticoagulation, steroid and ventilation may have also contributed to averting deaths[17,18]. In a sensitivity analysis (Supplementary Table 2 and Supplementary Fig. 6), we showed that adding the seroprevalence estimates obtained in[13] to our statistical framework did not change our projections and only affected the estimate of the probability of hospitalization upon infection $p_H$, which goes from 1.1% to 1.3%. Since seroreversion might have occurred for a proportion of those infected, we cannot exclude that the estimates of the serological study we used to validate our model might underestimate the real seroprevalence in French Guiana. However, since the vast

majority of infections occurred after April, i.e., less than 3 months before the survey was conducted, the impact of this phenomenon should have been relatively small at the time of the survey.

Major methodological developments have been made in the last few years to strengthen epidemic forecasting, with seasonal influenza or dengue constituting good case studies[19,20]. In a typical seasonal influenza epidemic, measures to reduce transmission in the general population are limited. As a consequence, once the epidemic has started, we expect that it will follow its natural course and that its trajectory can be forecasted if we have a good understanding of its key drivers (e.g., impact of the climate, population immunity, school holidays and circulating influenza subtype). In contrast, for SARS-CoV-2, unprecedented control measures are being implemented to limit spread; in addition, individuals are likely to naturally modify their behaviours (e.g., to reduce their contacts) as the pandemic progresses in their community[21]. A simple international comparison shows how the control measures and behaviours that are adopted can radically change the course of the pandemic from scenarios of near-suppression in South Korea and New Zealand to much less favourable ones in Brazil and the US. In addition, both control measures and individual behaviours may quickly change with the epidemiological situation, in a way that may be hard to anticipate. All these elements explain why it is much more challenging to forecast the trajectory of the SARS-CoV-2 pandemic wave than that of, for example, a seasonal influenza epidemic. Given these difficulties, we prefer to talk about scenario analysis rather than forecasts.

French Guiana constitutes an interesting case study where a combination of strict interventions including curfews and localized lockdown coincided with a substantial reduction in SARS-CoV-2 transmission. We need to build on these local experiences to progressively determine the optimal set of interventions required to contain SARS-CoV-2 pandemic waves.

## Methods

We used a deterministic mathematical model to describe the transmission of SARS-CoV-2 and subsequent disease progression in the population of French Guiana. The compartmental structure of the model closely followed our previous work[2]: upon infection, susceptible individuals enter a first latent compartment where they are not infectious, while a second exposed compartment is used to capture individuals who are infectious but not yet symptomatic. Once infected, individuals can develop severe disease and require hospital and/or ICU care. We used two versions of the model. A first version explicitly accounted for the age structure in the population. To describe contact patterns in the population of French Guiana, we used a contact matrix from Suriname[22], a neighbouring territory with similar population structure. We used the following age-groups: 0–9y, 10–19y, 20–29y, 30–39y, 40–49y, 50–59y, 60–69y, and 70y+. Since the matrix in[22] uses instead 5-year age groups, we merged neighbouring bins by taking their weighted—by population size—average. In order to accelerate computation and shorten the turnaround time of our analyses, we developed a second version of the model in which we no longer explicitly included the population age structure in the model. We instead relied on a single severity parameter, the average probability of hospitalization given infection $p_H$. Assuming that the probability of infection is proportional to the daily number of contacts within each age group ($C_i$ for age-group i)[2], this severity parameter can be estimated from the age-specific probability of hospitalization upon infection ($p_H^i$ for age-group i) and the age distribution of the target population as follows:

$$p_H = \frac{\sum_{\text{Age group } i} p_H{}^i \cdot C_i \cdot n_i}{\sum_{\text{Age group } i} C_i \cdot n_i} \tag{1}$$

where $n_i$ is the number of individuals aged i in the target population. Throughout this analysis, we considered young people to be half as infectious as adults, since they exhibit the lowest prevalence of infections and the lowest risk of severe outcome from COVID-19[23]. Results obtained with the full age-structured model under our final assumptions closely matched those obtained with the simpler version (Supplementary Fig. 7).

In order to capture trends in the epidemic trajectory following the strengthening of control measures in French Guiana, we modified the structure of our model for the analyses we performed at the beginning of July 2020: while our initial model (M1) had a single change point for the transmission rate, our final model (M2) had two change points for this parameter. Supplementary Table 1 summarizes the models' key parameters.

The simulations were seeded on April 21st 2020 with an initial number of infectious individuals—split into the exposed and infectious compartments proportionally to the time spent in each compartment—that was estimated jointly with the parameters in Supplementary Table 1.

We fitted our models to daily hospitalization count data extracted from the SI-VIC database, which stores data on COVID-19 patients hospitalized in public and private hospitals in metropolitan France and overseas French territories. The data were corrected for reporting delays as described in[2]. Note that the surveillance system in French Guiana is the same as the one used in Metropolitan France both in terms of data collection strategies and indicators used to monitor the epidemic. According to official statistics however, access to care differs: for example, in 2014, 79% of individuals living in French Guiana consulted a general practitioner compared to 85% in Metropolitan France[24].

The model parameters were estimated via Markov Chain Monte Carlo (MCMC) sampling assuming a Poisson observation process and using uniform, non-informative, priors. We relied on the Deviance Information Criterion (DIC) for model comparison and selection[25], with smaller DIC values indicating stronger support for the model.

The model was implemented in C++, while R (version 4.0.2) was used to summarize and display the results.

Additional details on the model are provided in the Supplementary Information.

**Reporting summary**. Further information on research design is available in the Nature Research Reporting Summary linked to this article.

## Data availability

The data that support the findings of this study are available at the address: https://gitlab.pasteur.fr/mmmi-pasteur/covid19-guiana.

## Code availability

The code used for the analyses is available at the address: https://gitlab.pasteur.fr/mmmi-pasteur/covid19-guiana.

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

## Acknowledgements

We acknowledge financial support from the Investissement d'Avenir program, the Laboratoire d'Excellence Integrative Biology of Emerging Infectious Diseases program (Grant ANR-10-LABX-62-IBEID), Sante Publique France, the INCEPTION project (PIA/ANR-16-CONV-0005) and European Union V.E.O and RECOVER projects.

## Author contributions

A.A., C.T.K., and S.C. conceived the study. A.A. implemented the methods. A.A. and C.T.K. performed the analyses with contribution from J.P., P.B., and N.L. T.S., M.N., F.D., A.S., C.F., H.S., and C.R. contributed to data collection and interpretation of the results. A.A., C.T.K., and S.C. wrote the manuscript. All authors contributed to paper revisions.

## Competing interests

The authors declare no competing interests.
