## [Peer Review File · Nature Communications]

REVIEWER COMMENTS

Reviewer #1 (Remarks to the Author):

This is an important piece of work showing how the COVID-19 has evolved in French Guiana, how modelling has been used to inform control and how demographics are likely to shape the average severity of infection. In general I am highly supportive of it being published in Nature Communications as much of emphasis and model fitting to date has focussed upon settings with older demographics such that this represents an important contribution to the literature. However, I have to confess I was quite surprised at how brief and undetailed the authors thought the methods section should be to the extent that I had to contact the editor to check I wasn't missing something – not sure I've seen a 37 line methods section (incl. SI) before! Examining the code shows the work has clearly been competently carried out but I would expect a published version to have full details of model specification (e.g. parameter values and their derivation), the construction of the likelihood (incl. equations) and fitting with focus on key quantities – for example in the phrase “The probability of ICU admission given hospitalisation was estimated at 15.7% [13.9%, 17.6%] and the time spent in ICU at 15.0 [13.1, 17.4] days” – I have no idea how these were estimated, why they were estimated differently from previous analyses (which are defined according to a vaguely described ‘early estimates from routine data’) so would ask that the authors provide more detail – both to ensure the work is fully reproducible and to help future researchers with the process of responding to outbreaks in real time.

Apart from this pretty major suggested revision I also wondered why the authors did not consider incorporating the seroprevalence estimates within the fitting to provide an empirical estimate of the % of individuals hospitalised in French Guiana, on top of using it to provide validation for the previous modelling – I think that number would be very useful, as would any insight into demographics of those hospitalised. I think that would add a lot to the analysis.

I would argue that some of the phrasing/framing of the analysis needs to be thought about quite carefully – in particular the definition of a ‘worst case’ scenario needs to be carefully delineated as the being in the context of fairly stringent interventions “schools, places of worship and movie theaters stayed closed, while restaurants and bars were 93 allowed to reopen but were limited to outdoor sitting for on-site dining” so this is not to be confused with an unmitigated epidemic – I might rephrase to something like a scenario with ‘no further action’ or some such. Similarly I think the future projections are highly contingent on these measures remaining in place? I agree with the authors sentiments that the future of any COVID epidemic is highly uncertain but given the projection of a 30% attack rate and the observation that they can be much higher than that (e.g. in areas of neighbouring countries) – I would recommend being explicit that these projections are contingent on control being maintained for the foreseeable future. Any light the authors can shed on the impact of further release of control measures would be wonderful but I appreciate this is very difficult to answer and can, in my opinion, be addressed in the discussion.

In general I wondered if there was any data on the relative surveillance strength in Guiana relative to mainland? How does access to care differ? Are testing rates similar? Are deaths outside of hospital included in statistics? Is their data on time between onset and test etc.?

Apart from that I had the following minor comments:

Line 108 – “implementation... adapted to area according to the epidemiological situation”

This sentence is quite vague – what were the local measures and how were they adapted?

Line 109-118 – As well as my comment about worst case scenarios this is quite repetitive of things already written– suggest to edit and merge into intro

Line 156 – I would be explicit here that the timing of the changepoint itself is estimated

Line 170- ‘projections relatively stable’ – in what sense are they stable? You mean incidence is declining?

Line 197-199 – this approach targets the private sphere where social distancing more likely to be relaxed?

Not sure I quite followed this comment? Could it be clarified?

Reviewer #2 (Remarks to the Author):

Review of: "Evaluating the impact of curfews and other measures on SARS-CoV-2 transmission in French Guiana"

By Andronico et al.

17 Oct 2020

In this paper, the authors detail their efforts to model the impact of interventions on the SARS-CoV-2 epidemic in French Guiana.

The paper is interesting, but I think the authors need to make their purpose clearer. It reads like a blow-by-blow account of how their modelling assumptions changed over time while they were advising the French Guianese government. This is very interesting and makes for good reading, but to me it's not really clear whether Nature Communications is the right forum for this, since I'm not sure there is much scientific novelty here. This is not to take away at all from the research. It's not easy to do this work in real time and I'm sure the authors' work had a huge impact on policy. It's just that, reading the manuscript, I'm relating to and sympathizing with what the authors did, but not identifying the scientific message.

I think the authors overstate a bit the novelty of the findings concerning the differing impact of COVID-19 in younger populations, but to be fair this is one of my main research areas at the moment so I might be a bit harder to surprise on this front than most. As someone who advises government on COVID-19 policy I sympathize completely with the excitement of how assumptions and recommendations change over time, but I'm not sure what the general reader is supposed to take home from figs 2-4 which seem to mainly emphasize how the modelling results changed from month to month. The authors have to either go into much more detail about the significance of these changes over time, or just present the latest and most credible picture of the epidemic in French Guiana.

If the authors are trying to show that a SARS-CoV-2 epidemic in a young population can be readily explained by age structure alone — which is a major unsolved question — i.e. if they are trying to show evidence that there is no special "X factor" in younger populations (usually LMIC) that explains why cases, hospitalisations and deaths have been so much lower in young populations than in old populations, then they need to push this right to the front and focus less on how their modelling changed over time—in my opinion.

Minor comments:

Introduction

"Furthermore, underlying delays between infection and hospital admission mean that the effect of the interventions on the healthcare system can only be confidently measured a few weeks after their implementation."

This ignores the possibility of using more rapid metrics, such as test positivity rates or symptom reporting apps. These are never perfect, so I agree with the thrust of the sentence, but it needs to be put into context.

"minimal set of interventions"

Nitpicking, but it's not about the minimal set of interventions, but the set of interventions that will optimize various costs to society.

Results

"under the assumption that children were half as infectious as adults"

This assumption by the authors is reasonable to me, since I think there is good evidence that the symptomatic rate increases with age and symptomatic individuals are more infectious than asymptomatic individuals, and also that behavioural patterns and contact rates among adults are more conducive to transmission than those among children, but the statement glosses over a rather complex phenomenon and so requires further justification in the text of this article. School closures, behavioural factors, relative infectiousness of symptomatic individuals vs asymptomatic individuals, etc need to at least be hinted towards here, I think, for the comprehensibility of the article.

"where a complete lockdown was maintained"

Why was this done? It's a point of interest that bears elaboration.

"on Saturday during the weekend"

Redundant phrasing? Surely all Saturdays fall on the weekend.

"The implementation and schedule of the curfews were adapted to each area according to the epidemiological situation"

To help interpretation of the findings, this statement needs to be greatly expanded. Were there specific thresholds for implementation of local measures? How did the measures vary geographically? What were the most stringent and least stringent schedules for curfews adopted? Etc.

"based on early estimates from local data"

More details on how these were estimated would be helpful.

"with the basic reproduction number increasing from 1.35 [1.26, 1.45] (before May 20th) to 1.78 [1.68, 1.88] (afterwards)."

To estimate R_0 from case data, you also need to specify the generation interval. Would be good to have a note about how this was estimated by this point in the manuscript.

"based on early estimates from local data"

Given the centrality of the high severity / low severity / baseline estimates to the results, this estimation procedure needs more detail. What data were used? How were these estimates made based on these data?

"while the peak number of daily ICU admissions was projected..."

The authors have not yet detailed the proportion of all hospital admissions that result in an ICU admission, which makes this statement hard to interpret.

"better DIC support"

This needs to be explained for non-specialists in the main text.

“Once sufficient time had passed”

Can the authors define and justify “sufficient time”?

Also, the M1 model is not that plausible to begin with (R increasing in a time period that includes transmission control measures). I’m not sure it’s necessary to include in the paper insofar as it seems more like a historical detail of the modelling that was done.

“Assuming a time-dependent assay sensitivity (0% during incubation, 30.3% up to 10 days after symptom onset, 75% between 10 and 20 days after symptom onset, and 93.8% afterwards) as per the distributor specification”

Seroprevalence studies from the UK, at least, based on Euroimmun show a clear decrease in seropositivity over time, e.g. a quantifiable rate of seroreversion. Was this taken into account?

“However, while a number of control measures remained in place in French Guiana after the lockdown, they were insufficient to stop an important epidemic rebound. This epidemic was likely facilitated by the proximity of Brazil, a country that has experienced a very important pandemic wave [16,17], notably in neighboring Amazonian states.”

I can’t agree with this conclusion, given that the authors’ modelling shows that R never went below 1. Why should we agree with the conclusion that Brazil is to blame for this?

“This shows that it is essential to account for the age structure of a population to properly evaluate the impact of SARS-CoV-2 on its healthcare system.”

I don’t think most researchers would view this as a surprising discrepancy at this point. The age dependence of SARS-CoV-2 transmission has been amply demonstrated.

“Given these difficulties, we prefer to talk about scenario analysis rather than forecasts.”

This is a bit of a non sequitur. See general comments above.

Methods

“We adjusted the contact matrix for age-groups 0-9y, 10-19y, 20-29y, 30-39y, 40-49y, 50-59y, 60-69y, 70y+ accounting for the population structure of French Guiana.”

It is crucial to provide more details here. There are many ways to adjust a contact matrix for different age structures, and they each involve making certain assumptions about how interpersonal contacts scale with population distributions which in turn need to be justified.

Figures

Fig 2 – minor point, but can you rearrange the legends in A/B so they match the height of each peak?

Fig 2 — C/D it would be more informative to include vertical intercepts for the date of the modelled interventions rather than the date of the analyses.

** Fig 3 – M1 not realistic

** Fig 4 – arrangement of figs in terms of analyses made on different dates not very interesting

Reviewer #1 (Remarks to the Author):

This is an important piece of work showing how the COVID-19 has evolved in French Guiana, how modelling has been used to inform control and how demographics are likely to shape the average severity of infection. In general I am highly supportive of it being published in Nature Communications as much of emphasis and model fitting to date has focussed upon settings with older demographics such that this represents an important contribution to the literature.

We thank the reviewer for their supportive comment and the very helpful comments below.

However, I have to confess I was quite surprised at how brief and undetailed the authors thought the methods section should be to the extent that I had to contact the editor to check I wasn't missing something – not sure I've seen a 37 line methods section (incl. SI) before! Examining the code shows the work has clearly been competently carried out but I would expect a published version to have full details of model specification (e.g. parameter values and their derivation), the construction of the likelihood (incl. equations) and fitting with focus on key quantities – for example in the phrase “The probability of ICU admission given hospitalisation was estimated at 15.7% [13.9%, 17.6%] and the time spent in ICU at 15.0 [13.1, 17.4] days” – I have no idea how these were estimated, why they were estimated differently from previous analyses (which are defined according to a vaguely described ‘early estimates from routine data’) so would ask that the authors provide more detail – both to ensure the work is fully reproducible and to help future researchers with the process of responding to outbreaks in real time.

We thank the reviewer for this important comment. Part of the reason why the method's section was brief was that we adapted a model for Metropolitan France that was thoroughly described in a previous publication (Salje et al, Science 2020). However, looking back at our manuscript, we fully agree that, even if we consider the additional details provided in Salje et al, some methodological details were not provided. In the revised SI, we now provide all the technical details that were previously missing, including the exact system of ordinary differential equations that we used and the construction of the likelihood. Additionally, Supplementary Table 1 now contains a description of all model parameters along with their estimates.

Apart from this pretty major suggested revision I also wondered why the authors did not consider incorporating the seroprevalence estimates within the fitting to provide an empirical estimate of the % of individuals hospitalised in French Guiana, on top of using it to provide validation for the previous modelling – I think that number would be very useful, as would any insight into demographics of those hospitalised. I think that would add a lot to the analysis.

This is a good point. Initially, we decided to keep it out of the analysis as a validation, and to be able to present some real-time assessments (at time points when serology was not available). We keep this approach in the revised manuscript but now also present a sensitivity analysis where we show that adding the seroprevalence estimates to our statistical framework does not change our projections: it

only affects the estimate of the probability of hospitalization upon infection, which goes from 1.1% to 1.3% (Supplementary Table 2 and Supplementary Figure 5).

Concerning the demographics of those hospitalized: the new Figure 4 shows the age distribution of hospitalized cases in French Guiana and Metropolitan France.

I would argue that some of the phrasing/framing of the analysis needs to be thought about quite carefully – in particular the definition of a ‘worst case’ scenario needs to be carefully delineated as the being in the context of fairly stringent interventions “schools, places of worship and movie theaters stayed closed, while restaurants and bars were 93 allowed to reopen but were limited to outdoor sitting for on-site dining” so this is not to be confused with an unmitigated epidemic – I might rephrase to something like a scenario with ‘no further action’ or some such. Similarly I think the future projections are highly contingent on these measures remaining in place? I agree with the authors sentiments that the future of any COVID epidemic is highly uncertain but given the projection of a 30% attack rate and the observation that they can be much higher than that (e.g. in areas of neighbouring countries) – I would recommend being explicit that these projections are contingent on control being maintained for the foreseeable future. Any light the authors can shed on the impact of further release of control measures would be wonderful but I appreciate this is very difficult to answer and can, in my opinion, be addressed in the discussion.

We agree with the reviewer’s warning and have rephrased “worst-case scenario” as a “pessimistic scenario wherein local transmission rates would remain unchanged despite the additional control measures”. For the projections, we have added that they were “contingent on control measures being maintained” and have extended the discussion about uncertainties regarding next stages of the epidemic.

In general I wondered if there was any data on the relative surveillance strength in Guiana relative to mainland? How does access to care differ? Are testing rates similar? Are deaths outside of hospital included in statistics? Is their data on time between onset and test etc.?

Concerning the differences in surveillance systems and access to care, we have added the following sentence in the Methods section: “Note that the surveillance system in French Guiana is the same as the one used in Metropolitan France both in terms of data collection strategies and indicators used to monitor the epidemic. According to official statistics however, access to care differs: for example, in 2014, 79% of individuals living in French Guiana consulted a general practitioner compared to 85% in Metropolitan France [26].”

For the testing rates: it is difficult to accurately compare them, since the testing strategy has changed over time according to the epidemic situation in each region. However, at present, they should be roughly similar.

For the deaths: deaths outside the hospitals are not included in the surveillance system used to monitor the epidemic, because data consolidation can take up to a few weeks.

Since, for our analyses, we do not use data on tests or deaths, we have not discussed these last two points in the article.

Apart from that I had the following minor comments:

Line 108 – “implementation... adapted to area according to the epidemiological situation”

This sentence is quite vague – what were the local measures and how were they adapted?

We agree with the reviewer and have changed that sentence as follows: “Overall, the implementation and schedule of the curfews were adapted to each area according to the epidemiological situation: the tightening (or easing) of the curfews and the closure of nonessential businesses were all decided at the municipality level - and without resorting to a fixed threshold - by monitoring the number of detected and hospitalized cases. The measures were taken following the progression of the epidemic, which started in the east, and it then progressed along the coast (Cayenne and Kourou) to finally reach the west of the region (St-Laurent-du-Maroni) (see Figure 1).”

Line 109-118 – As well as my comment about worst case scenarios this is quite repetitive of things already written– suggest to edit and merge into intro

In the revised manuscript, we have removed the repeated sentence.

Line 156 – I would be explicit here that the timing of the changepoint itself is estimated

We have added the following sentence in the Results section: “the timing of the additional change point was estimated by comparing the models’ DICs (see Methods).”

Line 170- ‘projections relatively stable’ – in what sense are they stable? You mean incidence is declining?

This sentence has now been removed, but we meant that recalibrating the model with new data provided projections that did not vary much.

Line 197-199 – this approach targets the private sphere where social distancing more likely to be relaxed? Not sure I quite followed this comment? Could it be clarified?

We have rephrased that sentence as follows: “This approach therefore targets social interactions among family members, friends or close acquaintances, where social distancing is more likely to be more lax.”

Reviewer #2 (Remarks to the Author):

Review of: “Evaluating the impact of curfews and other measures on SARS-CoV-2 transmission in French Guiana” By Andronico et al.

17 Oct 2020

In this paper, the authors detail their efforts to model the impact of interventions on the SARS-CoV-2 epidemic in French Guiana.

The paper is interesting, but I think the authors need to make their purpose clearer. It reads like a blow-by-blow account of how their modelling assumptions changed over time while they were advising the French Guianese government. This is very interesting and makes for good reading, but to me it’s not really clear whether Nature Communications is the right forum for this, since I’m not sure there is much scientific novelty here. This is not to take away at all from the research. It’s not easy to do this work in real time and I’m sure the authors’ work had a huge impact on policy. It’s just that, reading the manuscript, I’m relating to and sympathizing with what the authors did, but not identifying the scientific message.

I think the authors overstate a bit the novelty of the findings concerning the differing impact of COVID-19 in younger populations, but to be fair this is one of my main research areas at the moment so I might be a bit harder to surprise on this front than most. As someone who advises government on COVID-19 policy I sympathize completely with the excitement of how assumptions and recommendations change over time, but I’m not sure what the general reader is supposed to take home from figs 2-4 which seem to mainly emphasize how the modelling results changed from month to month. The authors have to either go into much more detail about the significance of these changes over time, or just present the latest and most credible picture of the epidemic in French Guiana.

If the authors are trying to show that a SARS-CoV-2 epidemic in a young population can be readily explained by age structure alone — which is a major unsolved question — i.e. if they are trying to show evidence that there is no special “X factor” in younger populations (usually LMIC) that explains why cases, hospitalisations and deaths have been so much lower in young populations than in old populations, then they need to push this right to the front and focus less on how their modelling changed over time—in my opinion.

We thank the referee for their constructive comments. We agree that the paper was providing both a description of real time assessments used to inform policy making as well as scientific insights on SARS-CoV-2 epidemiology, which constituted an unusual mix and format. In the revised manuscript we have reduced descriptions on how the modeling results changed over time, which is of lesser interest, and put more emphasis on our scientific results on the epidemiology of SARS-CoV-2 in French Guiana.

In terms of purpose and novelty, we have tried to clarify in the revised manuscript that we provide novel insights on three important aspects of the epidemiology of SARS-CoV-2 in French Guiana:

- We believe that the most important one is the evaluation of a control strategy based on curfews. France has now implemented two lockdowns to stop quickly growing epidemics of SARS-CoV-2. While effective, these interventions are extremely costly for society. It is essential**

that we design alternative strategies that may effectively reduce spread but with lesser economic and social impact. We believe that the way the epidemic was controlled in French Guiana is an interesting approach that proved effective and did not require a generalized lockdown. The description of the strategy and the model-based evaluation of its impact we provide can help policy makers to evaluate the benefits of such an approach. This is the first purpose and novelty we emphasize in our introduction. These insights are critical in the current context in France where an extended nationwide curfew starting at 6pm has just been decided; and we believe it will be of interest for many other countries as they try to identify effective control strategies that are not as costly as a lockdown.

- The second point is indeed the comparison of the impact of SARS-CoV-2 in populations of different ages. We agree with the referee that the general result that the impact of SARS-CoV-2 spread is smaller in younger populations is not surprising nor novel. However, we feel that our analysis goes beyond such simple comparison. Indeed, in international comparisons, it may be difficult to dissociate the impact of age from that of other covariates - for example, high income countries tend to have old populations and good healthcare systems while low income countries often have younger populations and more limited healthcare facilities. What is interesting here is that the healthcare system is similar in Metropolitan France and French Guiana, that are two parts of the same country. This allows a more straightforward comparison than would be possible if considering populations of different countries. In this context, it is striking to observe that our model was able to correctly estimate the seroprevalence in French Guiana in July (observed: 15.4%; estimated: 17.6%), simply by adjusting for population demographics.
- While this part has been shortened, it remains useful to show how modelling can be used to support policy making.

In practice, we have largely rewritten the introduction to reduce the part about real-time modelling and better emphasize the key novelties of the paper. We extended the final part of the Results section to focus more on the differences between the age structure of Metropolitan France and French Guiana and what this entails in terms of epidemic management. We have removed Figure 4A to 4D (which showed how the projections evolved from July to the end of August) and replaced them with two new panels. The first one compares the age distribution of hospitalized cases in Metropolitan France and French Guiana, while in the second one we show what the epidemic in French Guiana would have looked like, had it occurred in a department of Metropolitan France.

We have kept Figure 2 and 3 together with a shorter description of our initial assessments, since, as pointed out by Reviewer #1, it provides an interesting description of how modeling can be used to inform control policies. Using modeling to guide policy making is not a novelty, but the scale at which this has been the case with COVID-19 certainly is.

Minor comments:

Introduction

“Furthermore, underlying delays between infection and hospital admission mean that the effect of the interventions on the healthcare system can only be confidently measured a few weeks after their implementation.”

This ignores the possibility of using more rapid metrics, such as test positivity rates or symptom reporting apps. These are never perfect, so I agree with the thrust of the sentence, but it needs to be put into context.

We have removed the sentence as we now put less emphasize on real-time modelling, in particular in our introduction.

“minimal set of interventions”

Nitpicking, but it’s not about the minimal set of interventions, but the set of interventions that will optimize various costs to society.

We have changed that sentence as “We therefore critically need to determine from local experiences of epidemic management which set of interventions may be sufficient for the control of SARS-CoV-2 epidemic while having the lowest societal cost.”

Results

“under the assumption that children were half as infectious as adults”

This assumption by the authors is reasonable to me, since I think there is good evidence that the symptomatic rate increases with age and symptomatic individuals are more infectious than asymptomatic individuals, and also that behavioural patterns and contact rates among adults are more conducive to transmission than those among children, but the statement glosses over a rather complex phenomenon and so requires further justification in the text of this article. School closures, behavioural factors, relative infectiousness of symptomatic individuals vs asymptomatic individuals, etc need to at least be hinted towards here, I think, for the comprehensibility of the article.

We have expanded on this in the Methods section: “Throughout this analysis, we considered young people to be half as infectious as adults, since they exhibit the lowest prevalence of infections and the lowest risk of severe outcome from COVID-19 [23].”

“where a complete lockdown was maintained”

Why was this done? It’s a point of interest that bears elaboration.

We have rephrased that sentence as follows: “A curfew was established from 11PM to 5AM every day except in Saint Georges, a city located on the border with Brazil, where a complete lockdown was maintained in order to contain the number of imported cases from the neighbouring country.”

“on Saturday during the weekend”

Redundant phrasing? Surely all Saturdays fall on the weekend.

The measure starts on Saturday at 3pm for the whole week-end (i.e. including the whole Sunday). This has been clarified.

“The implementation and schedule of the curfews were adapted to each area according to the epidemiological situation”

To help interpretation of the findings, this statement needs to be greatly expanded. Were there specific thresholds for implementation of local measures? How did the measures vary geographically? What were the most stringent and least stringent schedules for curfews adopted? Etc.

We have expanded that sentence as follows: “Overall, the implementation and schedule of the curfews were adapted to each area according to the epidemiological situation: the tightening (or easing) of the curfews and the closure of nonessential businesses were all decided at the municipality level - and without resorting to a fixed threshold - by monitoring the number of detected and hospitalized cases. The measures were taken following the progression of the epidemic, which started in the east, and it then progressed along the coast (Cayenne and Kourou) to finally reach the west of the region (St-Laurent-du-Maroni) (see Figure 1).”

“based on early estimates from local data”

More details on how these were estimated would be helpful.

This has been clarified in the Methods section of the supplementary materials and in Supplementary Table 1.

“with the basic reproduction number increasing from 1.35 [1.26, 1.45] (before May 20th) to 1.78 [1.68, 1.88] (afterwards).”

To estimate R_0 from case data, you also need to specify the generation interval. Would be good to have a note about how this was estimated by this point in the manuscript.

In the revised SI, we now provide all the technical details that were previously missing, including the exact system of ordinary differential equations that we used and the relationship between R_0 , transmission rate, and infectious period.

“based on early estimates from local data”

Given the centrality of the high severity / low severity / baseline estimates to the results, this estimation procedure needs more detail. What data were used? How were these estimates made based on these data?

This has been clarified in the Methods section of the supplementary materials and in Supplementary Table 1.

“while the peak number of daily ICU admissions was projected...”

The authors have not yet detailed the proportion of all hospital admissions that result in an ICU admission, which makes this statement hard to interpret.

We have added the following sentence before discussing the peak of daily ICU admissions: “Both estimates of the probability of ICU given hospitalization and of the average duration of stay in ICU were obtained using the model described in [12] (see also the Supplementary Information and Supplementary Table 1).”

“better DIC support”

This needs to be explained for non-specialists in the main text.

We have added the following sentence in the Results section: “the timing of the additional change point was estimated by comparing the models’ DICs (see Methods).”

Additionally, the Methods section contains the following sentence: “We relied on the Deviance Information Criterion (DIC) for model comparison and selection [25], with smaller DIC values indicating stronger support for the model.”

“Once sufficient time had passed”

Can the authors define and justify “sufficient time”?

We have removed that sentence.

Also, the M1 model is not that plausible to begin with (R increasing in a time period that includes transmission control measures). I’m not sure it’s necessary to include in the paper insofar as it seems more like a historical detail of the modelling that was done.

Model M1 has in fact a constant R in the period when control measures were strengthened (i.e. R does not increase). We think that it is interesting to show it alongside model M2 since it represents the - rather pessimistic - scenario wherein the strengthening of control measures does not affect transmission rates.

“Assuming a time-dependent assay sensitivity (0% during incubation, 30.3% up to 10 days after symptom onset, 75% between 10 and 20 days after symptom onset, and 93.8% afterwards) as per the distributor specification”

Seroprevalence studies from the UK, at least, based on Euroimmun show a clear decrease in seropositivity over time, e.g. a quantifiable rate of seroreversion. Was this taken into account?

This is now addressed in the Discussion: “Since seroreversion might have occurred for a proportion of those infected, we cannot exclude that the estimates of the serological study we used to validate our model might underestimate the real seroprevalence in French Guiana. However, since the vast majority

of infections occurred after April, i.e. less than 3 months before the survey was conducted, the impact of this phenomenon should have been relatively small at the time of the survey.”

“However, while a number of control measures remained in place in French Guiana after the lockdown, they were insufficient to stop an important epidemic rebound. This epidemic was likely facilitated by the proximity of Brazil, a country that has experienced a very important pandemic wave [16,17], notably in neighboring Amazonian states.”

I can't agree with this conclusion, given that the authors' modelling shows that R never went below 1. Why should we agree with the conclusion that Brazil is to blame for this?

We have changed the phrasing as follows: “...they were insufficient to stop an important epidemic rebound, which coincided with the surge of cases observed in Brazil, a country that has experienced a very important pandemic wave [14,15]....”

“This shows that it is essential to account for the age structure of a population to properly evaluate the impact of SARS-CoV-2 on its healthcare system.”

I don't think most researchers would view this as a surprising discrepancy at this point. The age dependence of SARS-CoV-2 transmission has been amply demonstrated.

We do not claim that this is a key novelty of the paper. This has been clarified as follows: “This shows that, as previously documented [2,8,9], it is essential to account for the age structure of a population to properly evaluate the impact of SARS-CoV-2 on its healthcare system.”

“Given these difficulties, we prefer to talk about scenario analysis rather than forecasts.”

This is a bit of a non sequitur. See general comments above.

We think that, in light of the uncertainties about SARS-CoV-2 spread that we discuss in that section, it is appropriate to clarify that the analyses we made were projections under different scenarios - which could not be anticipated with confidence.

Methods

“We adjusted the contact matrix for age-groups 0-9y, 10-19y, 20-29y, 30-39y, 40-49y, 50-59y, 60-69y, 70y+ accounting for the population structure of French Guiana.”

It is crucial to provide more details here. There are many ways to adjust a contact matrix for different age structures, and they each involve making certain assumptions about how interpersonal contacts scale with population distributions which in turn need to be justified.

We agree with the reviewer: we just meant that instead of using 5-year age groups (as in Prem et al.) we used 10-year age groups. The sentence now reads: “We used the following age-groups: 0-9y, 10-19y, 20-29y, 30-39y, 40-49y, 50-59y, 60-69y, and 70y+. Since the matrix in [22] uses instead 5-year age groups, we merged neighbouring bins by taking their weighted - by population size - average.”

Figures

Fig 2 – minor point, but can you rearrange the legends in A/B so they match the height of each peak?

Done.

Fig 2 — C/D it would be more informative to include vertical intercepts for the date of the modelled interventions rather than the date of the analyses.

Done.

Fig 3 – M1 not realistic

Model M1 represents a scenario wherein the strengthening of control measures does not affect transmission rates. We think that it is interesting to show the projections obtained under this rather pessimistic scenario alongside those from (the more realistic) M2.

Fig 4 – arrangement of figs in terms of analyses made on different dates not very interesting

We have moved all the panels showing projections at different dates to the Supplements.

REVIEWERS' COMMENTS

Reviewer #1 (Remarks to the Author):

I believe the changes made represent a substantive improvement to an already important contribution to the literature so I recommend publication. I do however have one minor request and is to ask whether the authors can also display (fine if in the SI) the hospitalisation data by age but weighted by population size of the given age-demographic (per-capita contribution to all hospitalisations or some such) - I think that would be really useful to see and can guarantee it would help garner at least a couple more citations... Though if not possible it wouldn't affect my recommendation.

I congratulate the authors on their analysis.

Patrick Walker

Reviewer #2 (Remarks to the Author):

The authors have suitably addressed my concerns. This is an interesting paper which makes a novel contribution to the evidence base on control of SARS-CoV-2.

I feel that an apology is warranted to the authors for the terseness of the last two points in my review. These were initial notes to myself which I later expanded upon further up in the review. I left these in by mistake, and did not mean to be as rude as I probably came across!

Reviewer #1 (Remarks to the Author):

I believe the changes made represent a substantive improvement to an already important contribution to the literature so I recommend publication. I do however have one minor request and is to ask whether the authors can also display (fine if in the SI) the hospitalisation data by age but weighted by population size of the given age-demographic (per-capita contribution to all hospitalisations or some such) - I think that would be really useful to see and can guarantee it would help garner at least a couple more citations... Though if not possible it wouldn't affect my recommendation.

This is now shown in Supplementary Figure 5.

I congratulate the authors on their analysis.

We thank the reviewer for the kind comments.

Reviewer #2 (Remarks to the Author):

The authors have suitably addressed my concerns. This is an interesting paper which makes a novel contribution to the evidence base on control of SARS-CoV-2.

I feel that an apology is warranted to the authors for the terseness of the last two points in my review. These were initial notes to myself which I later expanded upon further up in the review. I left these in by mistake, and did not mean to be as rude as I probably came across!

We thank the reviewer for their comments.